# Glioma Stem-Like Cells Can Be Targeted in Boron Neutron Capture Therapy with Boronophenylalanine

**DOI:** 10.3390/cancers12103040

**Published:** 2020-10-19

**Authors:** Natsuko Kondo, Masaki Hikida, Mitsutoshi Nakada, Yoshinori Sakurai, Eishu Hirata, Satoshi Takeno, Minoru Suzuki

**Affiliations:** 1Institute for Integrated Radiation and Nuclear Science, Kyoto University, Osaka 590-0494, Japan; yosakura@rri.kyoto-u.ac.jp (Y.S.); s_takeno@kuhp.kyoto-u.ac.jp (S.T.); msuzuki@rri.kyoto-u.ac.jp (M.S.); 2Department of Life Science, Akita University, Akita 010-8502, Japan; hikida@gipc.akita-u.ac.jp; 3Department of Neurosurgery, Kanazawa University Graduate School of Medical Science, Kanazawa 920-8640, Japan; mnakada@med.kanazawa-u.ac.jp; 4Division of Tumor Cell Biology and Bioimaging, Cancer Research Institute of Kanazawa University, Kanazawa 920-1192, Japan; ehirata@staff.kanazawa-u.ac.jp; 5Nano Life Science Institute, Kanazawa University, Kanazawa 920-1192, Japan; 6Department of Radiation Oncology, Graduate School of Medicine, Kyoto University, Kyoto 606-8501, Japan

**Keywords:** boron neutron capture therapy, glioma stem cell, boronophenylalanine, mass cytometry

## Abstract

**Simple Summary:**

Glioblastoma is the most lethal tumor among all cancers with a median overall survival of 14 months and to develop new effective therapies is an urgent issue. Boron Neutron Capture Therapy (BNCT) is a unique radiation therapy that uses boron compounds and thermal neutrons. This study is aimed to investigate whether glioma stem cells, which is resistant to chemo-radiation therapy, take up a boron compound, p-boronophenylalanine (BPA) or not. Both in vitro and in vivo studies, more glioma stem like cells took up BPA compared with the differentiated glioma cells and indicated that BNCT can target to kill GSCs and be an effective therapy for malignant glioma.

**Abstract:**

As glioma stem cells are chemo- and radio-resistant, they could be the origins of recurrent malignant glioma. Boron neutron capture therapy (BNCT) is a tumor-selective particle radiation therapy. ^10^B(n,α)^7^Li capture reaction produces alpha particles whose short paths (5–9 µm) lead to selective killing of tumor cells. P-boronophenylalanine (BPA) is a chemical compound used in clinical trials for BNCT. Here, we used mass cytometry (Cytof) to investigate whether glioma stem-like cells (GSLCs) take up BPA or not. We used GSLCs, and cells differentiated from GSLCs (DCs) by fetal bovine serum. After exposure to BPA for 24 h at 25 ppm in 5% CO_2_ incubator, we immune-stained them with twenty stem cell markers, anti-Ki-67, anti-BPA and anti-CD98 (heterodimer that forms the large BPA transporter) antibodies and analyzed them with Cytof. The percentage of BPA^+^ or CD98^+^ cells with stem cell markers (Oct3/4, Nestin, SOX2, Musashi-1, PDGFRα, Notch2, Nanog, STAT3 and C-myc, among others) was 2–4 times larger among GSLCs than among DCs. Analyses of in vivo orthotopic tumor also indicated that 100% of SOX2^+^ or Nestin^+^ GSLCs were BPA^+^, whereas only 36.9% of glial fibrillary acidic protein (GFAP)^+^ DCs were BPA^+^. Therefore, GSLCs may take up BPA and could be targeted by BNCT.

## 1. Introduction

Glioblastoma (GBM) is the most lethal primary brain tumor, with a median overall survival of 14 months. It inevitably recurs, even after surgical resection followed by intensive chemo-radiation therapy [1]. We have used boron neutron capture therapy (BNCT) to treat recurrent or newly diagnosed high-grade glioma and our results show prolonged survival with BNCT compared with the historical control cohort [2]. BNCT is a form of tumor-selective particle radiation therapy consisting of two components [3]. First, a boron-10 (^10^B)-containing drug is administered to the patient to obtain a sufficient tumor ^10^B concentration, and second, the tumor is irradiated with epithermal neutrons. The resulting ^10^B(n,α)^7^Li capture reaction produces alpha particles whose short path (5–9 µm) selectively kills tumor cells while sparing adjacent normal tissues. The most commonly used boron compound in BNCT clinical trials for glioma, head and neck cancer, melanoma and other cancers is *p*-boronophenylalanine (BPA). BPA is an analog of an essential amino acids and phenylalanine, and is actively taken up by tumor cells through L-Type Amino Acid Transporter 1 (LAT1), a member of the system L family of heterodimeric, sodium-independent, amino acid transporters [4], which is overexpressed in various cancers [5]. LAT1 is co-expressed with CD98 (4F2 heavy chain, the obligate chaperone) as a heterodimeric complex on the plasma membrane and transports the neutral amino acid phenylalanine [5]. Using BNCT, we achieved better local control and survival benefit in patients with malignant glioma compared with the control cohort. We found that even patients with recurrent gliomas, including those with poor prognosis, non-GBM patients with a Karnofsky performance score of less than 80 and GBM patients over 50 years of age with corticosteroid use, had a median survival time of 9.1 months after BNCT [6] compared with 4.4 months for patients with same prognostic factors within the New Approaches to Brain Tumor Therapy Central Nervous System (CNS) Consortium who were in other phase I/II trials for systemic or local therapy for recurrent glioma [2]. However, even after tumor-selective BNCT, recurrence was inevitable, the most frequent cause of death was cerebrospinal fluid (CSF) dissemination [2,7].

The reasons for recurrence after BNCT have not been fully elucidated, but they may reflect tumor characteristics. For example, we found that CSF dissemination after BNCT occurs more frequently in the small cell subtype of *IDH1^R132H^* mutation-negative GBM [7]. Another potential reason may be the heterogeneous distribution of BPA in the tumor, which consists of heterogeneous clones. Previous preclinical studies have reported heterogeneous distribution of BPA inside the peripheral (thigh) tumor using a melanoma or squamous cell carcinoma mouse model and indicated a relationship between the uptake of BPA and cell proliferation [8,9]. However, using quantitative subcellular imaging with secondary ion mass spectrometry, other studies of BPA showed that T98 GBM mitotic cells contain a significantly lower amount of boron in comparison with interphase cells [10]. Furthermore, Detta and Cruickshank reported that the uptake of BPA was antagonized by pretreatment with phenylalanine or a specific inhibitor of LAT-1, and the number of LAT-1-expressing cells was three times higher than that of cells expressing proliferating cell number antigen (PCNA) in glioma patient tumor samples (71.5 ± 17.02% versus 23.8 ± 16.5%; *p* < 0.0001; *n* = 38 GBM and metastatic tumors) [11]. These results indicate that non-proliferating cells could also take up BPA through LAT1.

Recent studies have shown that glioma stem cells (GSCs), a small subpopulation of tumor cells, are responsible for tumor resistance to radiation and chemotherapy, and the stemness, quiescence and therapy resistance are maintained by GSC niches in the tumor microenvironment [12,13]. However, BPA uptake in GSCs is largely unknown. Therefore, in this study, we investigated whether BPA is taken up by GSCs using mass cytometry (in vitro) and a mouse orthotopic tumor model (in vivo). We established two patient-derived glioma stem-like cells (GSLCs, named no. one and no. two) and their differentiated cells. Here, we report the possibility of BPA uptake by GSLCs.

## 2. Results

### 2.1. Differentiation Was Induced to Patient-Derived GSLCs by Fetal Bovine Serum

We established two GSLC lines, no. one and two, and induced differentiation by exposure to medium containing 10% fetal bovine serum (FBS) for 24 h. We examined two differentiation markers, glial fibrillary acidic protein (GFAP) for astrocytes and neuron-specific beta-III tubulin (Tuj1) for neurons. The differentiation markers GFAP and Tuj1 were expressed at higher levels in differentiated cells compared with GSLCs (Figure 1a, b). In contrast, the expressions of GSC markers (Oct3/4, SOX2, Nestin, PDGFRα, Nanog and STAT3) were decreased in differentiated cells after exposure to 10% FBS medium. Musashi-1, CD133, CD49f, Notch2, CD44, CXCR4 and c-Myc were decreased only in no. two cells after exposure to 10% FBS medium. CD171 expression increased in no. one cells and CD144 showed no change in both cell lines after differentiation (Figure 1a,b).

### 2.2. Larger Percentages of GSLCs Take up BPA Compared with Differentiated Cells

The percentage of the total BPA-positive cells decreased in differentiated cells compared with GSLCs in both no. one and no. two cells (stem vs. differentiated cells: no. one, 56.0% vs. 25.7% and no. two, 35.8% vs. 21.5%) (Table 1).

In differentiated no. one cells, only 9.3% were BPA+/Oct3/4+ cells compared with 29.4% of no. one GSLCs. Similarly, lower percentages of no. one differentiated cells expressed both BPA and a stem cell marker than detected in no. one GSLCs. In contrast, 18.4% and 24.9% of differentiated cells positive for BPA were positive for GFAP and Tuj1, respectively, compared with 20.5% and 21.2% in GSLCs positive for BPA (Table 1).

In differentiated no. two cells, only 2.9% were BPA+/SOX2+ cells compared with 15.5% of no. two GSLCs. Similarly, lower percentages of no. two differentiated cells expressed both BPA and a stem cell marker than were seen among the no. two GSLCs. In no. two cells, 8.6% and 9.4% of differentiated cells positive for BPA were positive for GFAP and Tuj1, respectively, compared with 10.5% and 11.1% GSLCs positive for BPA (Table 1).

### 2.3. Larger Percentages of GSLCs Express CD98 Compared with Differentiated Cells

Next, we compared the membrane CD98 expression between GSLCs and differentiated cells as a potential marker for BPA uptake (Table 2). Among differentiated no. one cells, only 20.4% were CD98+/SOX2+ cells compared with 51.1% of no. one GSLCs. Similarly, lower percentages of no. one differentiated cells expressed both CD98 and a stem cell marker than detected in the no. one GSLCs.

Among differentiated no. two cells, only 2.2% were CD98+/SOX2+ cells compared with 20.9% of no. two GSLCs. Similarly, lower percentages of no. two differentiated cells expressed both CD98 and a stem cell marker than detected in no. two GSLCs.

### 2.4. Some Stem Markers Have a Stronger Impact on the Uptake of BPA in GSLCs than Proliferation Markers

The percentages of no. one cells positive for the proliferation marker Ki-67 in stem and differentiated cells were 69.0% and 49.8%, respectively (Figure 2a). These results indicate that differentiation caused reduced proliferation in this study. To investigate the relationship between proliferation, stemness and BPA uptake, we set Ki-67 and SOX2 as two parameters and compared the percentage of BPA^+^ cells among no. one GSLCs (Figure 2b.1). SOX2^+^/Ki-67^+^/BPA^+^ and SOX2^+^/Ki-67^−^/BPA^+^ cells comprised 57.9% and 53.9% of no. one GSLCs, respectively, while SOX2^−^/Ki-67^+^/BPA^+^ and SOX2^−^/Ki-67^−^/BPA^+^ cells comprised 37.1% and 33.8% of no. one GSLCs, respectively (Figure 2b.2–5). We further used PDGFRα as another stem cell parameter in no. one stem-like cells (Figure 2c.1). PDGFRα^+^/Ki-67^+^/BPA^+^ and PDGFRα^+^/Ki-67^−^/BPA^+^ cells made up 57.4% and 50.7% of no. one GSLCs, respectively, while PDGFRα^−^/Ki-67^+^/BPA^+^ and PDGFRα^−^/Ki-67^−^/BPA^+^/cells made up 39.0% and 34.1%, respectively (Figure 2c.2–5). The differences between the Ki-67+ and Ki-67- cell populations were <7%, whereas those between the SOX2^+^ and SOX2^−^ cell populations were >20% and those between PDGFRα^+^ and PDGFRα^-^ populations were >16%.

The percentages of no. two cells positive for the proliferation marker Ki-67 in stem and differentiated cells were 42.6% and 31.4%, respectively (Figure 2d). We set Ki-67 and SOX2 as two parameters and compared the percentage of BPA^+^ cells among no. two GSLCs (Figure 2e.1). SOX2^+^/Ki-67^+^/BPA^+^ and SOX2^+^/Ki-67^−^/BPA^+^ cells comprised 48.1% and 44.3% of no. two GSLCs, respectively, while SOX2^−^/Ki-67^+^/BPA^+^ and SOX2^-^/Ki-67^−^/BPA^+^ cells comprised 35.4% and 33.3%, respectively (Figure 2e.2–5). We used STAT3 as another stem cell parameter in no. two stem-like cells (Figure 2f.1). STAT3^+^/Ki-67^+^/BPA^+^ and STAT3^+^/Ki-67^−^/BPA^+^ cells made up 45.9% and 42.2% of no. two GSLCs, respectively, and STAT3^−^/Ki-67^+^/BPA^+^ and STAT3^−^/Ki-67^−^/BPA^+^ cells made up 33.6% and 31.6%, respectively (Figure 2.f.2–5). The differences between the Ki-67^+^ and Ki-67^−^ cell populations were <4%, whereas those between the SOX2^+^ and SOX2^−^ cell populations or STAT3^+^ and STAT3^−^ populations were >11%. These results may indicate stem markers may have a stronger impact on BPA uptake than proliferation markers.

### 2.5. GSLCs Take up BPA at Higher Rates than Differentiated Cells in Tumors of Xenograft Model

Immunohistochemistry of tumor samples from an orthotopic mouse model by double staining of BPA and a stem marker (SOX2 or Nestin) or differentiation marker, GFAP, revealed that all SOX2- or Nestin-positive GSLCs were also positive for BPA (Figure 3a,b). In contrast, BPA-positive cells accounted for only 36.9% (±18.5%) of all GFAP-positive cells (Figure 3c), summarized in Table 3. These results indicate that GSLCs may take up BPA more effectively than differentiated cells and were consistent with the results obtained in vitro.

## 3. Discussion

Although BNCT prolonged the median survival time of patients with recurrent GBM with poor prognostic classification compared with controls [2], recurrence after BNCT is a critical problem to overcome. GSCs are thought to be key contributing factors to recurrence after standard chemo-radiation therapy [12]. In tumors treated with BNCT, incomplete uptake of BPA inside the tumor may lead to recurrence. Thus far, contradictory results have been reported on the relationship between BPA uptake and tumor cell proliferation. Some studies showed a positive correlation [8,9] while other studies showed no [11] or negative [10] correlations of these two parameters. Whether GSCs take up BPA is not well known and was a focus of investigation in this study.

We have shown that approximately 2–4 times more GSLCs (Oct3/4^+^, SOX2^+^, Nestin^+^, Musashi-1^+^, CD133^+^, PDGFRα^+^, Notch2^+^, Nanog^+^, STAT3^+^, c-Myc^+^ and CD49f^+^ cells, among others) take up BPA compared with differentiated cells in the patient-derived no. one and no. two cell lines. In addition, more GSLCs positive for stem markers expressed CD98, which forms a heterodimeric complex with LAT1 on the plasma membrane, compared with differentiated cells. Our results show that compared with stem cells, differentiated cells were less proliferative, these findings are similar to the reports that retinoic acid-induced differentiation in glioma cells and tissue also inhibited proliferation [14,15]. In addition, differentiated cells showed reduced BPA-positive cells, suggesting that BPA uptake might correlate with proliferation. Our data also identified not only BPA^+^/Ki-67^+^/stem marker^+^ populations but also the comparable BPA^+^/Ki-67^−^/stem marker^+^ populations. One of the novel findings of our study is that in GSCs positive for stem cell markers (SOX2, PDGFRα or STAT3), BPA+ populations were more than 10% larger compared with GSCs negative for stem cell markers, regardless of Ki-67 positivity. Therefore, stemness may have a stronger impact on BPA uptake in GSCs than proliferation.

A previous study using different patient-derived GSCs and differentiated cells indicated that GSCs take up BPA at lower concentrations per 10^7^ cells than differentiated glioma cells, but the retention time in GSCs was longer than that of differentiated cells [16]. A selective boron uptake in the tumors, developed from differentiated glioma cells or GSCs, was observed in mouse xenografts and the concentrations of ^10^B showed no difference between the tumor bulks, regardless of whether the tumor developed from the differentiated glioma cells or GSCs [16]. In addition, differentiated glioma cells were more sensitive than GSCs to BNCT, and both cells underwent apoptosis through the mitochondria pathway [17]. In this study, we aimed to clarify the intratumor heterogeneity of BPA uptake in terms of stemness. We induced differentiation from two patients’ GSLCs in vitro and found that BPA uptake was more efficient in both sets of GSLCs compared with induced differentiated glioma cells. In addition, analysis using an orthotopic xenograft model also indicated that GSC marker-positive GSLCs more efficiently take up BPA than differentiated GFAP-positive glioma cells. Our results show that inside the tumor derived from cells from each patient, GSLCs more easily take up BPA than differentiated cells and can be targeted by BNCT.

LAT1 transports amino acids as a complex composed of LAT1 covalently bound to 4F2 heavy chain antigen (also called CD98, official gene name: *SLC3A2*). CD98 promotes LAT1 protein stability and mediates the translocation of LAT1 to the cell membrane [6]. In lung cancer tissues, the expressions of LAT1 and CD98 are significantly correlated [18]. In our experiments, BPA uptake and CD98 positivity were well correlated, which may help elucidate why more stem-like cells (Oct3/4^+^, SOX2^+^, Nestin^+^, Musashi-1^+^, CD133^+^, PDGFRα^+^, Notch2^+^, Nanog^+^, STAT3^+^, c-Myc^+^ and CD49f^+^ cells) tend to express CD98 and take up BPA compared with differentiated cells. Notably, SOX2 and c-Myc bind to most of the same promoters [19] and the *LAT1* promoter has a c-Myc binding sequence [20], which suggests that *SLC3A2* (the gene encoding CD98) and *LAT1* may be target genes of SOX2. SOX2 and Oct3/4 form a complex on the enhancer of genes related to the maintenance of pluripotent embryonic cells [21,22,23]. Therefore, SOX2 and Oct3/4 might form a complex and bind to the promoters of *LAT1* and *SLC3A2*. Previous studies have also reported a strong correlation between stemness and LAT1 or CD98. After retinoic acid treatment, NT2 cells differentiated into neurons with reduced LAT1 and CD98 expression [24]. The Notch pathway, which is also important for regulating GSC self-renewal and differentiation, was identified as being important for LAT1 expression in T-cell acute lymphoblastic leukemia cells [25]. In glioma, the functions of Enhancer of zeste homolog 2 (EZH2), which was not tested as a stem marker in this study, are closely related with stemness and mesenchymal transition [26]. The expressions of LAT1 and EZH2 have also been linked with more undifferentiated cancer. Culturing lung cancer cells as pulmospheres to promote their differentiation caused decreased levels of LAT1, CD98 and EZH2 proteins [27]. Within lung tumor samples, LAT1 and EZH2 were co-expressed in cancer cells that were more undifferentiated and highly proliferative, compared with adjacent stromal tissues that did not express these proteins [27].

A previous study that investigated the impact of CD98 on stemness, proliferation and cell survival in acute myelogenous leukemia showed that loss of CD98 triggers apoptosis and depletion of acute myelogenous leukemia stem cells and CD98-mediated adhesion to vasculature promotes leukemia stem cell maintenance [28]. The vascular niche is one of the important GSCs niches that sustain stemness and resistance to chemoradiation therapy [12,13]. We speculate that GSCs with CD98 expression that reside in the vascular niche can take up BPA through LAT1 in a heterodimeric complex with CD98 on the plasma membrane and can be targeted by BNCT as our data show. Therefore, vascular niches may play a critical role in the uptake of BPA by GSCs. Notably, although GSCs take up BPA and BNCT can target GSCs, peri-hypoxic niches, one of the GSC niches, can dedifferentiate glioma differentiated cells into GSCs [29], and this may contribute to the resistance to BNCT.

We confirmed that GSLCs treated with FBS expressed increased levels of differentiation markers (GFAP and Tuji-1). In contrast, FBS reduced the expression of stem markers (Oct3/4, SOX2, Nestin, Musashi-1, PDGFRα, Notch2, Nanog, STAT3, c-Myc and CD49f) in GSLCs. These results are consistent with retinoic-acid induced changes in differentiation and stem markers in GSLCs [13]. Among the 18 stem cell markers we tested, significant differences between GSLC and differentiated cells were observed in the expressions of Oct3/4, SOX2, Nestin, PDGFRα, Nanog and STAT3 in both no. one and no. two cells. In no. two cells, the expression of other stem cell markers (Musashi-1, CD133, CD49f, Notch2, CD44, CXCR4 and c-Myc) also differed between stem-like cells and differentiated cells. The 24 h treatment with FBS might have affected genes that code for regulatory proteins that function at early stages of lineage commitment (Oct3/4, SOX2, Nestin, PDGFRα, Nanog and STAT3), as shown in no. one and no. two cells. Cell surface stem cell markers, including cell adhesion and cell–cell/cell–matrix interactions, may increase at the second wave of gene regulation by FBS (which occurred only in no. two cells in this study) as shown in retinoic acid-induced regulation in embryonic carcinoma and embryonic stem cells [30,31]. In addition, the change in PDGFRα expression may have occurred earlier than for STAT3 and c-Myc in no. one GSLCs. PDGFR binds and activates signal transducers and is an activator of transcription (STATs). Phosphorylation of Y705 in STAT3 leads to dimerization, nuclear translocation, recognition of STAT3-specific DNA binding elements and up-regulation of various STAT3 downstream target genes, such as *Bcl-XL*, *Bcl-2*, *survivin*, *c-Myc* and *cyclin D1* [32].

## 4. Materials and Methods

### 4.1. Cell Lines

We used human GBM cell lines that were established from tumor samples from two patients and named the cell lines no. 1 and no. 2. To establish the cells, we minced and pipetted the tumor samples in TrypL Express (Thermo Fisher Scientific, Waltham, MA, USA) and incubated them in a CO_2_ incubator at 37 °C, cells were rinsed and collected after filtering through a 40-µm cell strainer (Thermo Fisher Scientific) to remove any aggregates. Use of human materials and protocols were approved by the Ethics Committees of Kanazawa University and Kyoto University (R0534). Cells were cultured as non-adherent spheroids in serum-free DMEM/F12 containing GlutaMax (Thermo Fisher Scientific), B27 without vitamin A (Thermo Fisher Scientific), penicillin and streptomycin (Nacalai Tesque, Kyoto, Japan), hEGF (20 ng/mL) and hFGF (20 ng/mL, Peprotech, TX, USA). To induce cell differentiation, we exposed the cells to media containing 10% FBS for 24 h. We dissociated cells to single cells using accutase (Nacalai Tesque) and rinsed them with PBS before mass cytometry analysis.

### 4.2. BPA Exposure

We treated cells with medium containing BPA at the concentration of 25 ppm for 24 h. The BPA was formulated and its concentration was measured as previously described [33].

### 4.3. Multiparameter Mass Cytometry

The panel comprised the following markers: for brain tumor stem cells, cell-surface, CD133 PE-conjugated (#130-098-826, Miltenyi Biotech, Bergisch Gladbach, Germany; anti-PE-^156^Gd, Fluidigm, South San Francisco, CA, USA, #3156005B), CD15 (#3144019C, ^144^Nd, Fluidigm), CD171 (#371602, ^147^Sm, Biolegend, San Diego, CA, USA), IL6Ra (#352801, ^149^Sm, Biolegend), CD144 (#348501, ^154^Sm, Biolegend), PDGFRα (#3160007C, ^160^Gd, Fluidigm), CD49f (#3164006C, ^164^Dy, Fluidigm), Notch2 (#3165026C, 165Ho, Fluidigm), CD44 (#130-099-091, ^166^Er, Miltenyi), EGFR (#3170009C, ^170^Er, Fluidigm), CXCR4 (#306501, ^175^Lu, Biolegend); intracellular, Oct3/4 (#sc-5279, ^146^Nd, Santa Cruz Biotechnology, Santa Cruz, CA, USA), SOX2 (#3150019C, ^150^Nd, Fluidigm), Nestin (#3151013C, ^151^Eu, Fluidigm), Musashi-1 (#3155013C, ^155^Gd, Fluidigm), Nanog (#3169014C, ^169^Tm, Fluidigm), STAT3 (#3173003C, ^173^Yb, Fluidigm) and c-Myc (#3176012C, ^176^Yb, Fluidigm); for differentiated cells: Tuj1 (#801201, ^141^Pr, Biolegend) and GFAP (#3143022C, ^143^Nd, Fluidigm); for proliferation markers: intracellular, Ki-67 (#130-100-340, ^168^Er, Miltenyi); for BPA uptake: intracellular, BPA (biotin-conjugated, anti-biotin-^171^Yb) and cell-surface, CD98 (#3159022C, ^159^Tb, Fluidigm). All metal-tagged antibodies were obtained from Fluidigm except the BPA antibody [34,35] (kindly provided by Professor M. Kirihata in Osaka Prefectural University) and CD133, CD171, IL6Ra, CD144, CD44, CXCR4, Oct3/4, Tuj1 and Ki-67 antibodies, which were conjugated in-house. Cells were resuspended in 1 mL of PBS and stained for 5 min with 1 µM cisplatin. After rinsing with PBS, cells were incubated with the cocktail of cell-surface antibodies (1:100 dilution) in 100 µL of PBS for 30 min at room temperature. After being washed with PBS, cells were fixed on ice in 200 µL fixation/permeabilization solution (#554714, BD Biosciences, CA, USA). After washing with Perm/Wash buffer (#554714, BD Biosciences), cells were incubated with the cocktail of intracellular antibodies (1:100 diluted) in 100 µL of PBS for 30 min on ice. After washing with Perm/Wash buffer, cells were incubated with 200 µL of Perm/Wash buffer containing 0.125 µM Intercalator-Ir (Fluidigm) for 15 min at room temperature. After being washed with Perm/Wash buffer and two washes with water, cells were resuspended in water. Samples were acquired on a cytometry time of flight (Cytof) system (Fluidigm) at an event rate of 300–500 event/s. Prior to analysis, data were gated to eliminate dead cells. The analysis of Flow Cytometry Standard (FCS) files was carried out using Kaluza software (Beckman Coulter, Miami, FL, USA).

### 4.4. Experimental Orthotopic Tumor Model

Following Institutional Animal Care Use Committees guidelines in an institutional review board-approved protocol (approval no. 39), we implanted dissociated no. 1 cells into two 10-week-old C57BL/6 nude mice (Japan SLC, Inc., Shizuoka, Japan). Throughout the experiments, mice were housed in a controlled environment, and food and water were available. Animals were anesthetized and immobilized in a small animal stereotactic device (Narishige, Japan). Cells (2 × 10^5^) were injected using a Hamilton syringe, through a hole 2 mm to the right of the bregma, at a depth of 3 mm, at a rate of 1 µL/min. The hole was sealed with bone wax, and the skin over the injection site was sutured. Animals were monitored daily and sacrificed on the first sign of neurological deficit. Two hours before sacrifice, BPA (500 mg/kg) was subcutaneously injected. The brain tissue was removed and fixed with 4% paraformaldehyde overnight, the samples were transferred to 20% and 30% sucrose in PBS and kept in the solution until they sank to the bottom. Thereafter, the tissue blocks were frozen and coronal sections (6 µm in thickness) were cut with a Leica cryostat.

### 4.5. Immunohistochemistry

Antigens were unmasked by 60 min of incubation in 70 °C citrate buffer (pH 6.0) and then samples were pre-incubated with a blocking solution (10% normal goat serum and 1% bovine serum albumin, 0.3% Triton X-100). For blocking of endogenous IgG, sections were incubated with an unconjugated affinity purified F(ab) fragment anti-mouse IgG (H+L) (#ab6668, Abcam, Cambridge, UK) overnight at 4 °C at 0.1 mg/mL. Samples were incubated with the following primary antibodies (1:200 dilution): mouse anti-BPA (provided by Professor Kirihata), rabbit anti-SOX2 (#ab97959, Abcam), mouse anti-human Nestin (#MAB5326, Millipore Sigma, MA, USA) and mouse anti-GFAP (#MS-1376-P0, Thermo Scientific, MA, USA). Slides were then washed several times with PBS and incubated with an Alexa Fluor 488 goat anti-mouse antibody, an Alexa Fluor 594 goat anti-mouse antibody or an Alexa Fluor 546 goat anti-rabbit antibody for 1 h at room temperature in the dark. When samples were stained with two mouse-derived primary antibodies, we treated sections by one primary antibody followed by the appropriate secondary antibody, and then repeated the same procedure, from blocking to another primary antibody and the following secondary antibody. The sections were then washed with tris-buffered saline containing Tween 20 and mounted with 50% glycerol with 4′,6-diamidino-2-phenylindole (Santa Cruz Biotechnology). Images were acquired using a BZ-9000 microscope (Keyence, Osaka, Japan) and digitally processed, and counting was performed with the BZ-X analyzer software (Keyence). In each section, three fields of ×200 magnification were used for counting. Values are presented as means ± standard deviation.

## 5. Conclusions

BNCT is a tumor-selective particle radiation therapy that uses BPA. Heterogeneity in BPA uptake is thought to be a contributing factor to recurrent tumors. Mass cytometry is a recent advanced single cell technology that enabled us to show that brain tumor stem-like cells can take up BPA with greater CD98 expression than the differentiated cells. Our results showing that GSCs take up BPA and can be targeted by BPA-based BNCT have important implications, as the usual X-ray radiation and chemotherapy are not effective for GSCs. The combination of BNCT and X-ray irradiation at the same time may result in better prognosis for glioma patients. Future studies should clarify the molecular profiles of tumor cells that do not take up BPA including differentiated cells and cannot cause ^10^B(n,α)^7^Li capture reaction. Tumor associated macrophages are also worth investigating for BPA uptake, since they constitute up to 30% of the tumor mass and support tumorigenesis and tumor expansion [36].

## Figures and Tables

**Figure 1 cancers-12-03040-f001:**
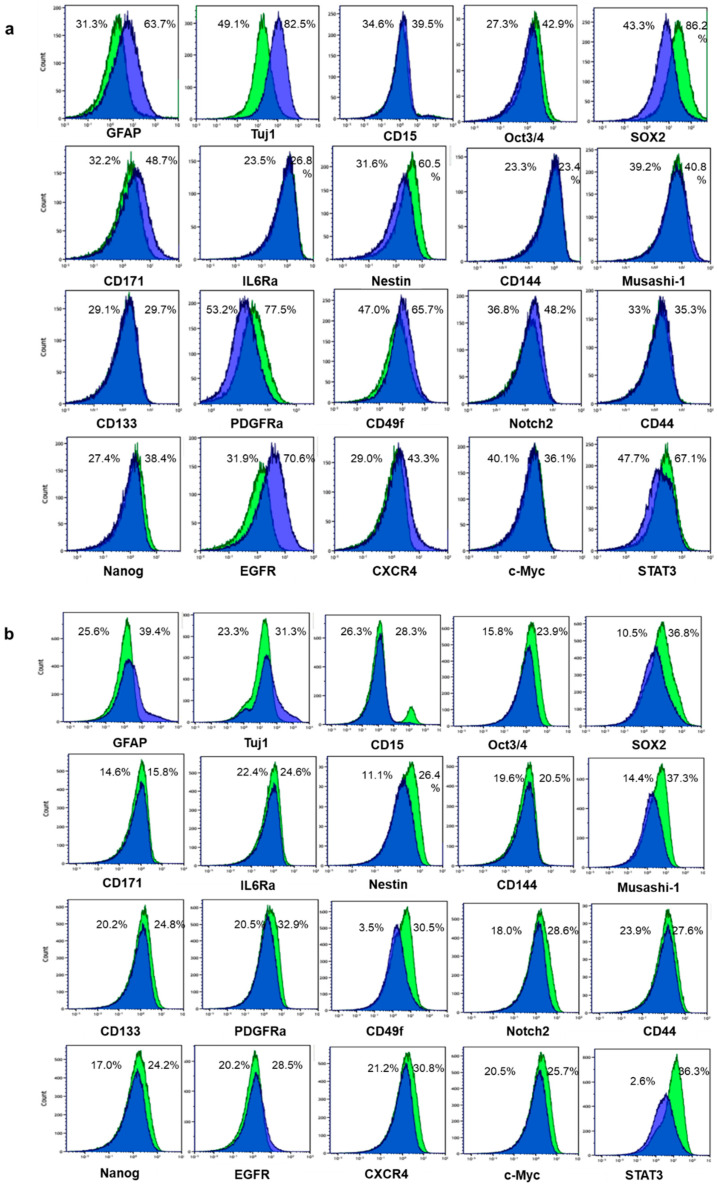
Differentiation was induced in patient-derived glioma stem-like cells (GSLCs) no. 1 (**a**) and no. 2 (**b**) by fetal bovine serum. Green histograms: GSLCs; blue histograms: differentiated cells. Changes in differentiation markers glial fibrillary acidic protein (GFAP) and neuron-specific beta-III tubulin (Tuj1) and stem cell markers are shown. Percentages of the positive cells for each indicated marker are shown.

**Figure 2 cancers-12-03040-f002:**
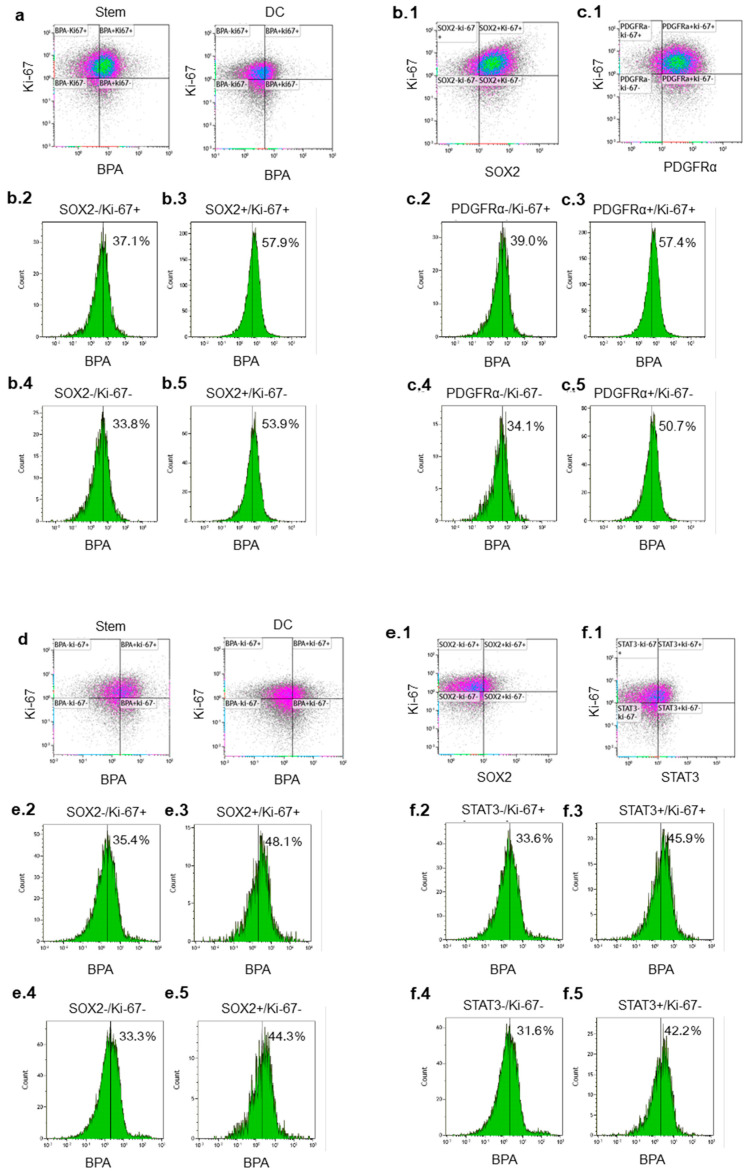
Relationships among proliferation marker Ki-67, stem markers and BPA uptake. (**a**) Total stem and differentiated cells (DC) from no. 1 cells and expressions of (**b.1**) SOX2 and (**c.1**) PDGFRα in no. 1 stem-like cells; (**d**) total stem and DC from no. 2 cells and expressions of (**e.1**) SOX2 and (**f.1**) STAT3 in no. 2 stem-like cells. Percentages in (**b.2**–**5**), (**c.2**–**5**), (**e.2**–**5**) and (**f.2**–**5**) indicate BPA^+^ cells for each stem marker^+^ or stem marker^−^/Ki-67^+^ or Ki-67^−^ populations.

**Figure 3 cancers-12-03040-f003:**
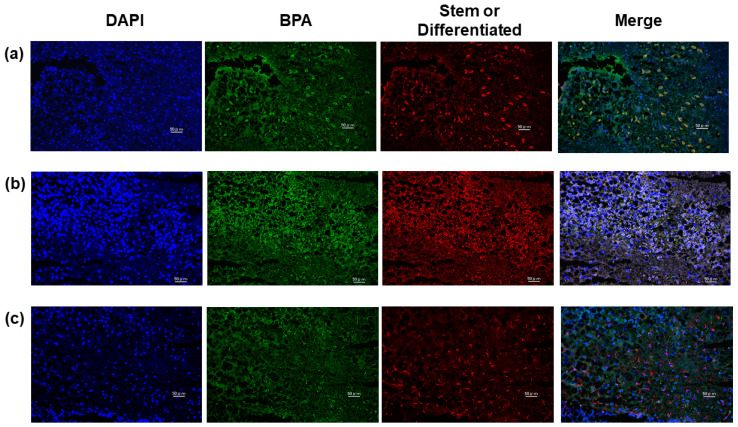
Uptake of BPA in tumors of an orthotopic mouse model. Tumor samples were co-stained with BPA and (**a**) SOX2, (**b**) Nestin or (**c**) glial fibrillary acidic protein (GFAP). Scale bar: 50 µm.

**Table 1 cancers-12-03040-t001:** Percentages of total *p*-boronophenylalanine (BPA^+^) cells, BPA^+^/differentiation marker^+^ cells and BPA^+^/stem marker^+^ cells among glioma stem-like cells (Stem) and differentiated cells (DC) from no. 1 and no. 2.

BPA^+^	No. 1 Stem	No. 1 DC	No. 2 Stem	No. 2 DC
Total	56.0	25.7	35.8	21.5
**Differentiation ^+^**				
GFAP	20.5	18.4	10.5	8.6
Tuj1	21.2	24.9	11.1	9.4
**Stem ^+^**				
Oct3/4	29.4	9.3	11.4	4.9
CD15	19.2	10.6	11.0	6.1
CD171	21.3	13.9	8.0	4.2
IL6Ra	15.1	6.4	8.1	4.1
SOX2	51.9	15.2	15.5	2.9
Nestin	35.0	8.3	12.8	3.6
CD144	11.6	5.3	6.8	3.9
Musashi-1	28.1	14.5	19.0	8.2
CD133	18.8	9.0	11.6	5.8
PDGFRα	44.9	15.6	12.0	4.6
Notch2	18.9	12.1	11.8	4.6
CD44	19.1	9.9	11.6	5.9
Nanog	20.6	6.9	11.5	5.0
STAT3	44.9	17.2	16.2	1.1
CXCR4	18.7	12.2	13.5	5.7
c-Myc	27.1	11.6	13.2	5.7
CD49f	29.8	18.7	15.0	2.0

**Table 2 cancers-12-03040-t002:** Percentages of CD98^+^/stem marker^+^ cells among glioma stem-like cells (Stem) and differentiated cells (DC) from no. 1 and no. 2.

CD98^+^	No. 1 Stem	No. 1 DC	No. 2 Stem	No. 2 DC
Stem^+^				
Oct3/4	29.2	13.3	15.6	3.6
CD15	19.3	14.8	16.1	4.9
CD171	23.0	22.0	11.1	3.1
IL6Ra	15.1	9.1	11.8	3.5
SOX2	51.1	20.4	20.9	2.2
Nestin	34.8	11.8	18.3	3.0
CD144	11.7	7.8	9.3	3.0
Musashi-1	27.5	20.0	21.4	2.9
CD133	19.2	12.9	16.5	4.7
PDGFRa	47.7	23.2	18.2	3.9
Notch2	18.3	16.6	17.2	3.5
Nanog	20.3	9.5	15.9	3.8
STAT3	43.5	23.7	23.6	0.8
CXCR4	21.1	22.8	19.7	4.4
c-Myc	27.7	17.5	18.4	4.6
CD49f	29.4	26.5	22.8	1.6

**Table 3 cancers-12-03040-t003:** Percentages of stem marker (SOX2 or Nestin)^+^ or differentiated marker (GFAP)^+^/BPA^+^ cells among glioma cells in tumors of orthotopic xenograft developed from no. 1 glioma stem cells (GSCs).

Stem or Differentiated Marker	Co-Expression of BPA (%)
SOX2	100
Nestin	100
GFAP	36.9 (±18.5)

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
