# Peer review of "Glioma Stem-Like Cells Can Be Targeted in Boron Neutron Capture Therapy with Boronophenylalanine"

_cancers, 2020, doi:10.3390/cancers12103040_

Round 1
Reviewer 1 Report
The authors have satisfactorily incorporated my suggestions in their manuscript.
Reviewer 2 Report
Glioblastoma is a very lethal, and the most common, brain tumor with poor patient survival of approximately 15 months. The poor survival of glioblastoma patients is attributed to the presence of glioma stem cells (GSCs). GSCs are generally quiescent very cells, which are very resistant to both radiotherapy and chemotherapy. To find improved treatments modalities of Glioblastoma is therefore of great importance so the paper is of interest for the community.
I have reviewed the paper again and taken part of previous review reports and in general I think the responses from the authors to the reviewers comments are satisfactory, so the paper can be accepted for publication.
This manuscript is a resubmission of an earlier submission. The following is a list of the peer review reports and author responses from that submission.
Round 1
Reviewer 1 Report
The authors present a potentially interesting in vitro study on the specificity of boron neutron capture therapy (BCNT) for glioma stem-like cells. While the presented data is of potential interest, several issues and unclarities preclude publication at this point:
The authors describe BNCT as achieving a survival benefit in patients with recurrent glioma. Although the peer-reviewers and editors of their original, and valuable, J Neurooncol publication included this "Survival Benefit" in their publication title, it seems to definitive to claim that BNCT achieves a survival benefit without a proper randomized clinical trial. It would be more appropriate to more explicitly state that BNCT was associated with a prolonged survival when compared to a historical RPA-matched control cohort.
The 3+7 RPA class in the introduction should be either stated more generally or explained more explicitly, because it is not a common term for reader that are not familiar with the specifics of this study.
It seems unnecessary to abbreviate median survival time to MST when it only occurs twice in the entire manuscript.
In the introduction, the authors describe that BNCT accumulates preferentially in growing cells rather than in quiescent tumor cells. Based on this statement, it seems unlogical to hypothesize why BNCT would be an effective therapy to target GSCs, since GSCs are generally quiescent cells that reseed tumor recurrences while more differentiated glioma cells express more proliferation and are responsible for the growth of the tumor bulk (Hira et al, Biochim Biophys Acta Rev Cancer. 2018 Apr;1869(2):346-354), https://www.ncbi.nlm.nih.gov/pmc/articles/PMC3097894/, https://www.frontiersin.org/articles/10.3389/fncel.2018.00388/full, http://genesdev.cshlp.org/content/29/12/1203.full.html, http://cancerres.aacrjournals.org/content/71/3/634.long. The introduction could be improved by elaborating on the hypothesis why BNCT would be specific for and a good therapy against BNCT to not confuse readers with this apparent contradiction.
The tumor microenvironment, or GSC niches should be mentioned in the introduction, as niches maintain GSC stemness, quiescence and therapy-resistance.
The nomenclature "KGS01" and "KGS03" in the Results section is unclear to readers not familiar with these cell lines, especially since the journal style places the Methods section at the end of the manuscript. The readability of the Results section would be improved by specifying the origin of these cell lines and perhaps renaming them to cell lines with numbers 1 and 2, since it is now unclear to readers why KGS02 is missing.
Explain in the results that GFAP is a marker for astrocytes and TUJ-1 a marker for neurons. "Differentiation markers"is too general.
In the results in Figure 1, CD171, CD144 and STAT3 are not mentioned/explained in the text.
It is highly recommended to include percentages in the histograms in Figures 1a and 1b, as the right shift in the histograms is not always clear and convincing. For example, in Fig. 1b, GFAP and TUJ-1 expression do not seem different in GSCs vs differentiated KGS03 glioblastoma cells (especially if compared to GFAP and TUJ-1 expression in KGS01 cells where a clear right shift is shown in Fig 1a). Does this mean that the KGSO3 cells did not differentiate after exposure to FBS?
In Table 1, the combinations of GFAP+BPA+ cells and TUJ-1+BPA+ cells should be included, to convince the readers that in the GFAP and TUJ-1-positive differentiated cells, BPA uptake was lower than in GSCs.
In Results section 2.4, the authors sum up the percentage of BPA+ cells based on the SOX2, Ki67, PDGFRA status. For example: "Sox2+/Ki-67+/BPA+ and Sox2+/Ki-67−/BPA+ cells comprised 57.9% and 53.9% respectively." Throughout the manuscript, the differences between the Ki-67+ and Ki-67- cell populations are <7%. Therefore, the statement from the Introduction section that "accumulates preferentially in growing cells rather than in quiescent tumor cells". The mansucript would be improved by comparing the novel data with those presented in the referenced publication (Int. J. Radiat. Oncol. Biol. Phys. 1996, 34, 1081-1086; DOI:0360301695021809) to provide more context to the readers.
One line below, a "/" is probably missing from "Sox2−/Ki-67+ BPA+" and "Sox2−/Ki-67− BPA+"
In the Discussion, mention how/whether GSC niches affect BPA uptake by GSCs and their resistance to BNCT.
In the Discussion section, a word may be missing from "Detta and Cruickshank found LAT1+ cells percentage was approximately 72.6%"
One paragraph below, the authors write "To our knowledge, our findings provide the first evidence that GSLCs take up BPA and can be targeted by BNCT." To support this statement, the authors should provide context and comparison as to why their findings are the first evidence that GSLCs take up BPA and can be targeted by BNCT, in relationship to previous publications on this matter: https://www.ncbi.nlm.nih.gov/pubmed/27191269 and https://www.ncbi.nlm.nih.gov/pubmed/23915425 and https://www.ncbi.nlm.nih.gov/pubmed/22728842.
It seems unnecessary to abbreviate antibodies to Abs when it only occurs thrice in the entire Methods section.
Include a Table in the Methods section with the source (company, catalog numbers) of all antibodies and the dilutions that were used and indicate for each marker whether it is a cell surface marker (no fixation and permeabilization) or an intracellular marker (requires fixation and permeabilization).
Since the cell lines used in the manuscript are of human nature, human protein nomenclature (SOX2) should be used instead of murine protein nomenclature (Sox2).
Reviewer 2 Report
The authors investigated the uptake of bronophenylalanine (BPA) in glioma stem like cells (GSLCs) for the possibility of targeting them in boron neutron capture therapy (BNCT). They directly compared patient-derived GSLC cell lines to their differentiated counterparts for the amount of BPA uptake and CD98 downstream transporter marker. They used CyTOF mass spectrometry to characterize a panel of stem cell and differentiation markers and revealed that GSLCs showed higher uptake of BPA and higher expression of CD98 compared to the differentiated cells. With this, the authors concluded that GSLCs uptake BPA and could be targeted by BNCT.
Major comments:
(1) the authors did not address the rationale behind using BPA, given that BNCT using BPA for glioma does not achieve efficacy in the clinic (regardless of the mechanism).
(2) The authors should provide direct evidence for difference in the total BPA uptake in stem-like cells vs the differentiated cells.
(3) The authors should provide direct evidence for difference in the proliferation (using Ki67) in stem-like cells vs the differentiated cells.
(4) In the last figure, all the cells seem to be highly proliferative and highly stem-like (third quadrant); therefore, the claim of “stemness had a stronger impact on BPA uptake than did proliferation” is not supported by evidence. The authors should present flow-cytometry figures with Ki67 in the x-axis and BPA uptake in the y-axis with clear separation between two distinct population of proliferating and non-proliferating cells (for both the stem- like and the differentiated cells).
(5) The gating for Ki67 and Sox2 is so different for KGS01 and KGS01 cells, which makes the results less conclusive, especially with the lack of isotype controls.
(6) As the main limitation of the use of BPA for BNCT has been the low T/B ratio and low T/N ratio, it is highly important to perform in vivo bio-distribution experiments to show, at least, BPA levels in stem cell like-tumor cells, differentiated-tumor cell, blood, and normal brain.
Minor Comments:
(1) More introduction about the clinical reasons for the failure of BPA in clinical setting is needed. As well as discussion of how this paper may improve the previous limitations.
(2) The data presentation in the figure is confusing, especially in the presentation of the tables, and the lack of isotype-controls in the last figure.
(3) English level is poor; grammar and sentence structures need substantial improvement.